# M-LongDoc: A Benchmark For Multimodal Super-Long Document Understanding And A Retrieval-Aware Tuning Framework

## Abstract

The ability to understand and answer questions over documents can be useful in many business and practical applications. However, documents often contain lengthy and diverse multimodal contents such as texts, figures, and tables, which are very time-consuming for humans to read thoroughly. Hence, there is an urgent need to develop effective and automated methods to aid humans in this task. In this work, we introduce M-LongDoc, a benchmark of 851 samples, and an automated framework to evaluate the performance of large multimodal models. We further propose a retrieval-aware tuning approach for efficient and effective multimodal document reading. Compared to existing works, our benchmark consists of more recent and lengthy documents with hundreds of pages, while also requiring open-ended solutions and not just extractive answers. To our knowledge, our training framework is the first to directly address the retrieval setting for multimodal long documents. To enable tuning open-source models, we construct a training corpus in a fully automatic manner for the question-answering task over such documents. Experiments show that our tuning approach achieves a relative improvement of 4.6% for the correctness of model responses, compared to the baseline open-source models. [1]

## 1 Introduction

The ability to comprehend long and complex multi-modal documents and respond to user queries about them is crucial in various practical applications such as business intelligence analysis, academic literature review, and legal research (Mathew et al., 2020). Recently, large multimodal models such as GPT-4V (OpenAI, 2023) have shown great potential in processing and analyzing diverse types of information, including text, images, and even structured data (Huang et al., 2024b). These models offer the promise of automating tasks that traditionally required extensive human effort, such as document analysis, information retrieval, and question-answering (Fujitake, 2024). However, real-world documents often present significant challenges due to their length, complexity, and multimodal nature, containing a mix of text, figures, tables, and charts (Faysse et al., 2024). Thus, it is not clear whether current models are capable of an in-depth understanding of lengthy multimodal documents. On the other hand, while existing benchmarks have fostered great progress in document understanding, they often fall short in representing these challenges, typically focusing on documents with less than 50 pages, and limited to simpler extraction-based questions (Ma et al., 2024).

To address these limitations, we introduce M-LongDoc, a comprehensive benchmark consisting of 851 samples specifically designed to evaluate the performance of large multimodal models on lengthy and diverse documents. Unlike previous datasets (Mathew et al., 2020; Liu et al., 2024; Ma et al., 2024) that mainly contain short documents, M-LongDoc features recent documents *spanning hundreds of pages, encompassing a wide range of topics and document structures* as shown in Figures 1 and 2. In addition, as shown in Figure 3, our benchmark *goes beyond simpler extractive questions, requiring models to provide open-ended solutions* that demonstrate in-depth understand-

---

[1]We will make the M-LongDoc benchmark, the training corpus for multimodal document reading, and our source code publicly available.

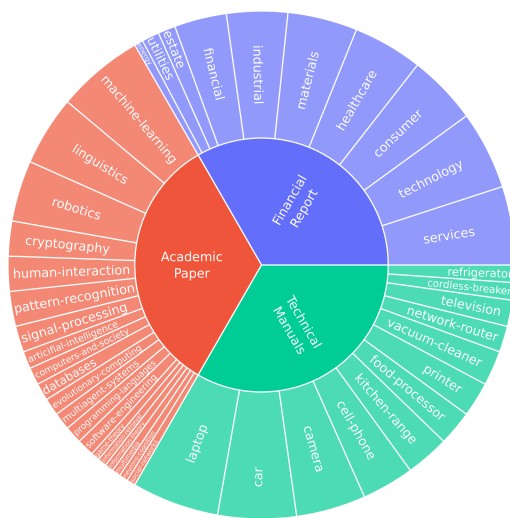

Figure 1: Data distribution of document topics in our M-LongDoc benchmark.

|  | Pages | Tokens | In-Depth |
|---|---|---|---|
| DocVQA | 1.0 | 151.5 | ✗ |
| ChartQA | 1.0 | 236.9 | ✗ |
| InfoVQA | 1.2 | 288.0 | ✗ |
| TAT-DQA | 1.1 | 577.0 | ✗ |
| VisualWebBench | 1.0 | 452.4 | ✗ |
| PWC | 12 | 7000 | ✗ |
| MP-DocVQA | 8.3 | 2026.6 | ✗ |
| DUDE | 5.7 | 1831.5 | ✗ |
| SlideVQA | 20.0 | 151.5 | ✗ |
| MMLongBench | 47.5 | 2030.5 | ✗ |
| **Ours** | 210.8 | 120988.0 | ✓ |

Figure 2: Comparison of benchmarks along three dimensions: the number of pages per document, the number of tokens per document, and the nature of the responses required. Specifically, we assess whether each benchmark emphasizes in-depth, comprehensive answers or focuses on short or extractive responses.

ing of the document content (Fan et al., 2019). M-LongDoc poses a question answering task where models have to analyze and reason over texts, figures, or tables in each multimodal long document. We aim for this benchmark to serve as a valuable resource for researchers and practitioners, enabling more rigorous testing and development of multimodal document understanding systems.

Another challenge we have to overcome is that due to the lengthy content of multimodal documents and the in-depth solutions, the evaluation of the open-ended question-answering task becomes tricky. To assess such open-ended solutions in a scalable and standardized manner, we design an automated evaluation framework that does not require reference answers or human annotation. Inspired by previous works in model-based evaluation (Zheng et al., 2023; Zhao et al., 2024; Liu et al., 2023c), our evaluation framework leverages a detailed evaluation guide and multiple judge models to score the correctness of each generated solution.

With our proposed M-LongDoc and evaluation framework, we conducted preliminary study on existing models and the results show that they struggle with figure and table-based questions compared to text-based questions, revealing their multimodal bias and weaknesses (Chen et al., 2024b). Furthermore, we observed that the models can be easily distracted by irrelevant content in the document pages (Shi et al., 2023), even with the aid of retrieval-augmented generation. To enhance the robustness of multimodal models against potentially irrelevant retrieved content, we propose a retrieval-aware tuning approach for multimodal document reading. This framework unifies supervised fine-tuning and retrieval augmented generation by including distracting content from other modalities and pages in each document. Thus, we adapt models to effectively incorporate the domain knowledge in multimodal documents while ignoring the content irrelevant to the given query. Compared to existing training methods, ours is the first to address retrieval-augmented multimodal training for rich document layouts. To support this training framework and the enhancement of open-source models, we further construct a training corpus of 10,070 samples for question-answering over multimodal long documents. Experiments show that our approach achieves a 4.6% relative improvement in the correctness of model responses, compared to the baseline model.

The key contributions of this work are threefold: 1) We introduce M-LongDoc, a multimodal benchmark that more accurately represents the challenges of real-world document understanding tasks. Our automated evaluation framework enables scalable and standardized assessment of open-ended solutions. 2) Our evaluation of leading models indicates that most models struggle with figure and table-based questions compared to text-based questions, revealing their multimodal bias. 3) We propose a retrieval-aware tuning framework that together with our large-scale training corpus, significantly improves the efficiency and effectiveness of multimodal document reading.

| Dataset | Example Question | Example Answer |
|---------|------------------|----------------|
| DocVQA | What is the underlined heading just above the table? | Indications for implantation |
| | What is the Extension Number as per the voucher? | (910) 741-0673 |
| MMLongBench | What is the number of red logos in page 10? | 0 |
| | I'm at the location "J" shown in the campus map. Tell me the name of the nearest coffee shop. | Ten Years After Cafe |
| Ours |  Where are the oven vents located on this range model, and why is their positioning important for proper oven function? | The oven vents are located at the top front of the oven, with one vent on the upper front and another on the lower front. Their positioning is important for proper oven function because they release hot air and moisture from the oven during cooking and cleaning. Blocking or covering the vents can cause poor air circulation, affecting cooking and cleaning results. The vents also help to maintain a consistent temperature in the oven by releasing excess heat and preventing the oven from overheating. |

Figure 3: Example questions in different multimodal document question answering benchmarks. For illustration, we include content from the relevant page in the original document. The example question from M-LongDoc is more complex than those from other benchmarks, as it requires an explanatory answer rather than an extraction of a short text span. Furthermore, it requires the model to understand the semantics of both image and text. Please note that in our benchmark setting, the model is provided with all page contents from the document, and not only the relevant page.

Thus, we believe that this work contributes to the field of document understanding and paves the way for more capable and practical applications of large multimodal models in real-world scenarios. To accelerate the studies in our community, we will make the M-LongDoc benchmark, the training corpus for multimodal document reading, and our source code publicly available.

## 2 M-LONGDOC BENCHMARK

To evaluate the multimodal long document understanding ability of existing models, we present M-LongDoc, a challenging and diverse benchmark. Notably, the benchmark focuses on open-ended questions that require in-depth solutions and analysis over very long documents with more than 200 pages on average. For diversity, the questions cover the academic, financial, and product domains, with multiple topics in each domain.

### 2.1 DATA COLLECTION

To support our evaluation benchmark, we manually source high-quality multimodal documents from publicly accessible sources. Concretely, we source research papers[2], company reports[3] and product instruction manuals[4] for the academic, financial, and product domains respectively. Thus, the dataset covers a range of document formats and domains. As research papers often require domain expertise, we constrain the academic domain to computer science topics. To reduce the risk of data contamination or memorization when evaluating existing models (Dong et al., 2024), we constrain the documents to be published in January 2024 or later. As most existing models are unable to process raw PDF files, we conduct a simple data processing to extract the texts and relevant images from each document. Specifically, we use the PyMuPDF[5] tool to automatically extract the text from each page. To extract the figures and tables from each page, we leverage an existing object detection model (Pfitzmann et al., 2022). Thus, the processed documents consist of interleaved textual and visual content, where the visual contents are extracted images of figures and tables.

---

[2] https://arxiv.org
[3] https://www.annualreports.com
[4] https://www.manualslib.com
[5] https://pymupdf.readthedocs.io

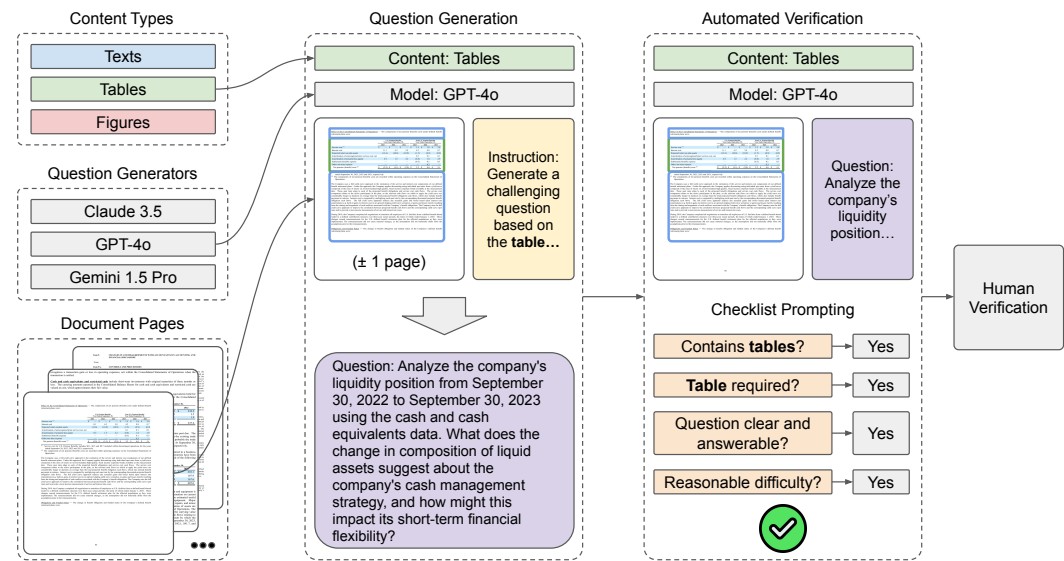

Figure 4: Overview of our data construction process with question verification stages. For brevity, we shorten the checklist prompts and include the full details in Appendix A.1.

## 2.2 QUESTION GENERATION

To construct diverse and challenging open-ended questions, we leverage a semi-automated pipeline. Concretely, as shown in Figure 4, given a specified content category, we first randomly select a page from the document that contains the specific content category, such as texts, tables, or figures. Consequently, we randomly select a question generator from a pool of leading multimodal models and instruct it to generate a challenging question based on the document page. To ensure that the question generator has sufficient context, we also provide the previous page and subsequent page as additional inputs during the question generation process.

To improve the quality of the generated questions, we conduct an automated verification process as a preliminary filter for unsuitable questions. Concretely, the question generator is also instructed to reflect on the generated question and follow a multi-step checklist to validate the question. For example, the checklist includes checking if the question is relevant to the document page, if the specified content category is required to answer the question, and whether the question is answerable. The question is rejected if it does not satisfy any condition in the checklist. Lastly, we employ a team of annotators to conduct final validation for each question. We employ expert annotators who are Ph.D. students and above in computer science for the academic domain, and professional annotators for the finance and product domains. To be consistent, we provide a similar checklist and instruction as our automated verification stage, and the annotation details are included in Appendix A.1. We found that 80.1% of the generated questions satisfied the automated verification. Of these questions that passed automated verification, 80.9% also satisfied the human verification. Thus, we only retain 851 questions that satisfied both the automated and human verification.

The statistics of our benchmark dataset are shown in Table 1, where we ensure a balanced distribution of questions and documents from each domain and question category. In this work, we focus on questions that require a single page of content to answer, and only retain answerable questions. Please also note that while each question focuses on a specific category in a document page, the page may contain content from other categories as context. For instance, a table-based question may also require comparisons to other tables or texts from the same page.

Compared to the existing benchmarks in Figure 2, M-LongDoc poses a greater challenge in two main aspects. Firstly, the significantly greater number of pages and tokens in each multimodal document poses extreme computational costs and opportunities to be distracted by irrelevant content (Shi et al., 2023). While this challenge may be mitigated by retrieval-augmented generation (Chen et al., 2022), our preliminary study in Section 2.4 shows that existing models are still hindered by their multimodal

|  | Academic Paper | Product Manuals | Financial Report | All |
|---|---|---|---|---|
| Documents | 60 | 60 | 60 | 180 |
| Questions | 311 | 279 | 261 | 851 |
|    Text-based questions | 95 | 95 | 81 | 271 |
|    Figure-based questions | 114 | 93 | 76 | 283 |
|    Table-based questions | 102 | 91 | 104 | 297 |
| Average pages per document | 201.2 | 277.8 | 153.4 | 210.8 |
| Average text tokens per document | 114,129.8 | 109,745.0 | 139,089.3 | 120,988.0 |
| Average figure images per document | 90.8 | 368.3 | 24.1 | 161.13 |
| Average table images per document | 34.9 | 96.6 | 83.8 | 71.8 |

Table 1: Benchmark dataset statistics with respect to each domain.

bias (Chen et al., 2024b). In addition, our benchmark poses challenging open-ended questions as shown in Figure 3, requiring models to produce in-depth analyses in their solutions. Thus, we believe M-LongDoc is a more realistic and challenging benchmark compared with existing datasets focusing on short answers that can often be extracted directly from the source document.

## 2.3 AUTOMATED EVALUATION

Given the challenging nature of our multimodal long document benchmark, it is crucial to have a scalable and standardized evaluation method. However, it is less feasible to conduct comprehensive human evaluation due to high labour costs and lack of reproducibility (Clark et al., 2021). Thus, inspired by previous works in automatic evaluation (Zheng et al., 2023; Zhao et al., 2024; Liu et al., 2023c), we propose an evaluation framework based on a committee of multimodal judges. Concretely, we leverage multiple leading multimodal models to score each answer to a question based on the criteria of correctness. To provide a clear guideline for evaluation, we define the task introduction and criteria as shown in Figure 5. To provide more detailed evaluation instruction beyond the basic definitions above, we further construct detailed evaluation steps, based on the task and criteria. Thus, the finalized evaluation guide for each judge model consists of the task introduction, criteria, and fixed evaluation steps.

To provide a more reliable evaluation and reduce intra-model bias (Verga et al., 2024), we leverage multiple judges to evaluate each candidate answer. Specifically, each judge model $M_j$ is provided with the evaluation guide $g$, ground-truth evidence page as context $c$, question $q$, and candidate answer $\hat{a}$, and instructed to assign a correctness score from 1 to 5. However, we observe some variance in the output scores, even with the same judge model and inputs. Thus, we sample multiple scores from each judge model $M_j$ and aggregate the scores to obtain a fine-grained, continuous score that better reflects the quality of the candidate answer:

$$\text{Score} = \frac{1}{J \cdot K} \sum_{j=1}^{J} \sum_{k=1}^{K} s_{j,k} \sim M_j(g, c, q, \hat{a}) \tag{1}$$

where $J = 3$ is the number of judge models and $K = 5$ is the number of sampled scores per judge model. While there may be some degree of subjectiveness in our framework, our analysis in later sections shows that it largely agrees with human preferences with minimal bias (Zheng et al., 2023). Thus, we believe this automated evaluation framework is reliable and more scalable.

## 2.4 PRELIMINARY STUDY

To investigate the limitations of existing models, we conduct a preliminary study on a subset of 100 random samples from our M-LongDoc benchmark. Concretely, we select Gemini (Google, 2024) and Qwen2-VL (Wang et al., 2024) to represent highly capable models for the close-source and open-source settings respectively. While large multimodal models have shown impressive capabilities and support longer input contexts, they often struggle with understanding very long documents and may incur great computational costs (Dingjie et al., 2024). Thus, we focus our study on the retrieval-augmented generation paradigm (Lewis et al., 2020), which leverages a retriever to select

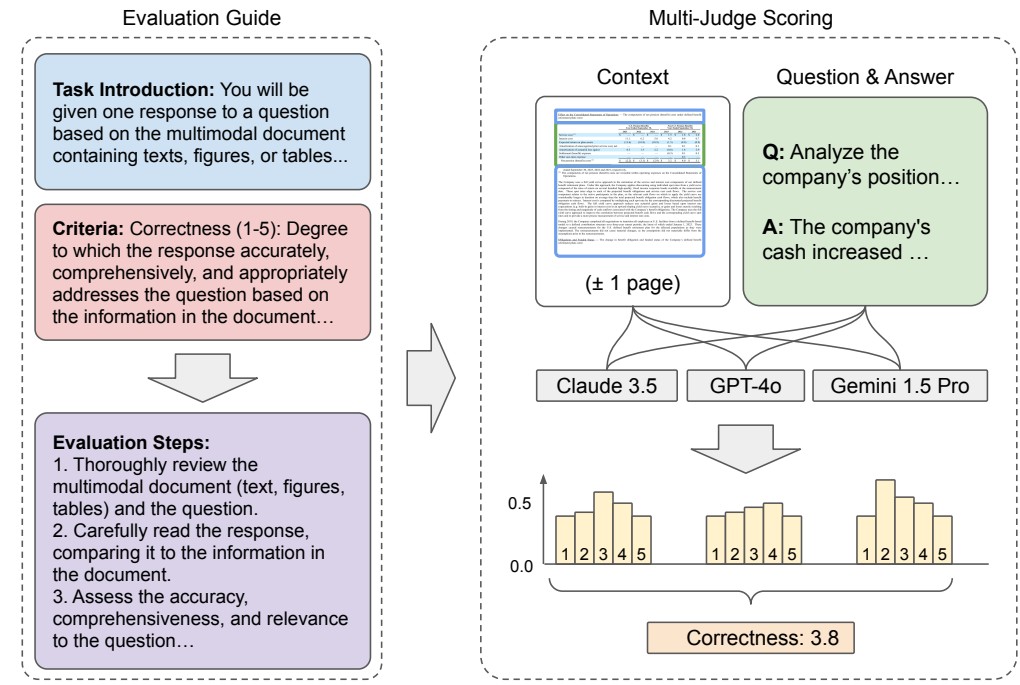

Figure 5: Our automated evaluation framework to assess the correctness of open-ended solutions for multimodal question answering. The full evaluation guide is included in Appendix A.3.

|  | Text | Figure | Table | All |
|---|---|---|---|---|
| Gemini-1.5-pro-002 |  |  |  |  |
| w/ top $k = 1$ pages | 4.38 | 3.73 | 4.16 | 4.11 |
| w/ top $k = 5$ pages | 4.60 | 4.31 | 4.54 | 4.49 |
| w/ top $k = 10$ pages | 4.61 | 4.29 | 4.62 | 4.51 |
| w/ top $k = 20$ pages | 4.63 | 4.33 | 4.38 | 4.46 |
| Qwen2-VL-7B-Instruct |  |  |  |  |
| w/ top $k = 1$ pages | 4.05 | 3.25 | 3.36 | 3.57 |
| w/ top $k = 5$ pages | 4.17 | 3.67 | 3.46 | 3.78 |
| w/ top $k = 10$ pages | 4.08 | 3.62 | 3.19 | 3.65 |
| w/ top $k = 20$ pages | OOM | OOM | OOM | OOM |

Table 2: Preliminary study on M-LongDoc for open-source and close-source models. We report the correctness score out of 5 for text-based, figure-based, table-based, and all questions respectively.

only the most relevant content, and the retrieved content is used to augment the generator model inputs for question answering. Concretely, we use ColPali (Faysse et al., 2024) as a state-of-the-art multimodal retriever and leverage the top $k$ pages of multimodal content as context. We include more details on the implementation and evaluation of retriever performance in Appendix A.2.

Notably, as shown in Table 2, we observe significantly lower performance for figure-based and table-based questions, as compared to text-based questions. We believe that this discrepancy suggests that current models are weaker in processing image-based contents in multimodal documents, or may be biased towards the textual content, even when they are trained on interleaved multimodal data (Chen et al., 2024b). Furthermore, we find that increasing the amount of retrieved content may not improve overall performance, and may even lead to worse performance or out-of-memory (OOM) issues. This indicates that the multimodal generator models may be easily distracted by irrelevant content in longer contexts (Shi et al., 2023). Thus, we believe it is crucial to address these challenges of existing models for processing multimodal long documents.

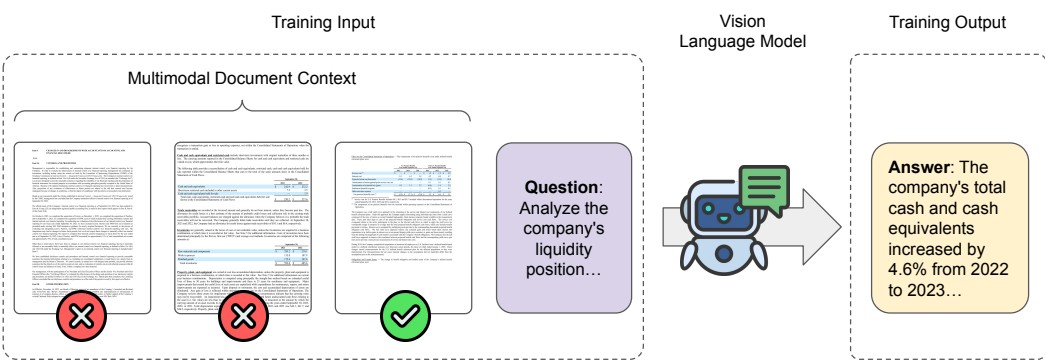

Figure 6: Our retrieval-aware multimodal tuning framework to enhance the ability of models to identify and utilize pertinent content in multimodal documents. At training time, the model is provided with more relevant pages retrieved from the document, which may contain both the gold evidence page and multiple 'distractor' pages.

Additionally, to verify the reliability of our automated evaluation, we conduct manual human scoring based on the same evaluation guide. For the samples in this preliminary study, we observed a Pearson correlation of 88.9% with $p < 0.001$ between the final aggregated score from the judge models, and the human annotator. Thus, we believe that our evaluation framework can achieve a very high agreement with human preferences despite the open-ended and in-depth nature of the answers.

## 3 RETRIEVAL-AWARE MULTIMODAL TUNING

While current multimodal models are often trained on interleaved inputs with multiple pieces of texts and images (Liu et al., 2023b), they may not be well-optimized for multimodal documents. Specifically, multimodal documents are more challenging to understand as they contain diverse multimodal content including unstructured texts, and images representing structured tables and figures. Thus, models require a more fine-grained understanding and perception of the multimodal document content. Furthermore, the retrieval setting requires models to ground their outputs in the additional context by identifying and utilizing only the pertinent content. However, previous works (Shi et al., 2023) and our preliminary study have shown that they may still be easily misled by some irrelevant information in the retrieved content. To our knowledge, there is no open-source model that can address these challenges in multimodal long documents.

To this end, we propose a simple and effective retrieval-aware multimodal document tuning approach. Inspired by previous works in retrieval augmentation (Chen et al., 2022; Zhang et al., 2024), we include both retrieval context from the ground-truth evidence page as well as potentially irrelevant pages during training. Our approach as shown in Figure 6 presents a training paradigm that is more realistic and similar to the challenges faced during test-time retrieval of multimodal content. Thus, the model learns to handle potentially noisy retrieval contexts, while improving its text understanding and visual perception capabilities to utilize the most relevant document content. To construct the training data, we leverage the same process as shown in Figure 4 to construct a training corpus of 10,070 samples across 300 documents, and leverage the respective question generator models to also produce a high-quality answer based on each ground-truth evidence page. We omit the human verification stage for scalability and cost-efficiency, as majority of the automatically verified samples also satisfied human verification. To assess the quality of the generated training solutions, we evaluated using our automated scoring framework on a random subset of 100 samples. We observed an average correctness score of 4.82, indicating very high quality of answers in the training data. To avoid data leakage, we ensure that the documents used to construct the training corpus do not overlap with the evaluation set. For example, we ensure that the training documents are from different companies and products, and are published in an earlier time period.

| Model | Size | Domain | | | Question Category | | | |
|---|---|---|---|---|---|---|---|---|
| | | Academic | Product | Finance | Text | Figure | Table | All |
| *Proprietary Models* | | | | | | | | |
| GPT-4o | - | 4.56 | 4.38 | 4.51 | 4.55 | 4.38 | 4.53 | 4.49 |
| Claude 3.5 Sonnet | - | 4.59 | 4.43 | 4.51 | 4.57 | 4.42 | 4.54 | 4.51 |
| Gemini 1.5 Pro | - | 4.66 | 4.43 | 4.43 | 4.59 | 4.43 | 4.52 | 4.51 |
| *Open-Source Models* | | | | | | | | |
| LLaVA OneVision | 7B | 3.71 | 3.74 | 3.39 | 4.03 | 3.57 | 3.30 | 3.62 |
| Qwen2-VL | 7B | 4.03 | 3.88 | 3.56 | 4.08 | 3.83 | 3.62 | 3.84 |
| w/ Retrieval Tuning | 7B | **4.17** | **4.01** | **3.86** | **4.31** | **4.00** | **3.77** | **4.02** |

Table 3: Evaluation of model performance for proprietary and open-source multimodal models. We report the correctness on our benchmark across different document domains and question categories. We bold the highest scores obtained by open-source models.

## 4 EXPERIMENTS

### 4.1 TASK SETTING

To ensure a practical task setting, we focus on the retrieval-based paradigm, which avoids the exorbitant cost to process the full document. Based on our preliminary study in Section 2.4, we use the top $k = 5$ pages ranked by the retriever as a reasonable amount of context for each question. Thus, each model is provided with the retrieved context and question as input, and required to provide an open-ended solution as output. As discussed in Section 2.3, we leverage an automated framework with multiple judge models to score the correctness of each output solution, on a scale of 1 to 5.

### 4.2 MODELS

To provide a more comprehensive investigation of current models, we use both open-source and close-source models in this work. Concretely, we select GPT-4o (gpt-4o-2024-05-13)[6], Claude 3.5 Sonnet (claude-3-5-sonnet-20240620)[7] and Gemini 1.5 Pro (gemini-1.5-pro-002) (Google, 2024) due to their leading performance on multimodal benchmarks (Yue et al., 2023). Regarding open-source models, we specifically select models which support interleaved multimodal inputs with multiple images, and fine-grained visual perception of document content. Thus, we mainly focus on LLaVA-OneVision-7B (Li et al., 2024) and Qwen2-VL-7B-Instruct (Wang et al., 2024). We plan to expand our investigation to other capable open-source models as they are released.

### 4.3 HYPERPARAMETERS

For all models, we use greedy decoding with temperature $T = 0$ to reduce variance. In our training framework, we set the number of training epochs to be 1, batch size as 16, and learning rate as 1e-4. To reduce the training cost due to limited computational resources, we leverage LoRA (Hu et al., 2022) training with rank as 64 and alpha as 32. Due to training instabilities with other open-source models, we mainly focus the training experiments on the Qwen2-VL-7B-Instruct model, which demonstrates leading performance compared to similar-sized models.

## 5 RESULTS

### 5.1 MAIN RESULTS

To assess the effectiveness of our approach and the holistic performance of existing models, we report the main evaluation results in Table 3. First, we find that our retrieval-aware multimodal

---

[6]https://openai.com/index/gpt-4o-system-card/
[7]https://www.anthropic.com/news/claude-3-5-sonnet

| | Question Category | | |
| Model | Text | Figure | Table |
| --- | --- | --- | --- |
| Qwen2-VL | 4.08 | 3.83 | 3.62 |
| w/o Image Inputs | 4.22 | 3.37 | 3.38 |
| w/ Render Page as Inputs | 3.99 | 3.70 | 3.39 |

Table 4: Analysis on alternative settings for our benchmark, including removing images from model inputs, and using only the render image of each page as document context, without text extraction.

tuning significantly and consistently enhances the performance of Qwen2-VL, representing a relative improvement of 4.6% in answer correctness. Thus, we view the proposed training approach as a promising strategy to enhance multimodal long document understanding ability, and reduce the gap between open-source and proprietary models. Second, we observe that open-source models have worse performance in answering table-related questions compared to other question categories. This discrepancy highlights the need for more efforts to enhance the table understanding capability of open-source multimodal models. We include further qualitative analysis of the model predictions in the Appendix A.4.

## 5.2 EFFECT OF ALTERNATIVE SETTINGS

While we mainly focus on the multimodal setting with extracted texts and images, we believe it is also important to explore other settings in practice. Our main data setting as discussed in Section 2.1 first extracts the texts, figures, and tables separately, with the figures and tables represented as individual images. As shown in Table 4, we find a significant decrease in performance of 12.0% and 6.6% respectively for figure-based and table-based questions when the image inputs are removed. However, the model may still be able to answer the questions to a limited extent as the extracted text may contain partial information about the tables and figures. We believe that this underscores the importance of leveraging multimodal content in documents, even though many documents may contain a majority of the content as texts. The performance increases slightly for text-based questions when image inputs are removed, suggesting that the images may mislead the model in rare cases. However, we believe this is acceptable as questions may cover a wide variety of multimodal content in practice. On the other hand, we observe that it may be less optimal to use only rendered images of document pages as inputs, instead of separately extracting the texts, tables, and figures as in our main setting. While the rendered page image does contain the original information and layout of the document, including texts, tables, and figures, the model may be less capable of distinguishing the content between texts and tables.

## 6 RELATED WORK

### 6.1 LARGE MULTIMODAL MODELS

In recent years, large multimodal models have demonstrated their capability to process and comprehend data across various formats. Close-source models such as GPT-4o (AI, 2024) can reason across audio, vision and text. Claude 3.5 Sonnet (Anthropic, 2024) shows marked improvement on tasks that require visual reasoning like interpreting charts and graphs. On the other hand, Gemini 1.5 Pro (Google, 2024) is capable of reasoning over multiple long documents and hours of video and audio. Open-source models such as Llava (Liu et al., 2023a), Idefics (Laurençon et al., 2023), Otter (Li et al., 2023), InternVL (Chen et al., 2024c), CogVLM (Wang et al., 2023), have also shown the potential over various types of multi-modal content including document images (Mathew et al., 2020), slides (Tanaka et al., 2023), and charts (Huang et al., 2024a). However, the benchmark performance of open-source models tends to lag behind that of close-source models (Yue et al., 2023), prompting an urgent need to bridge the gap. In this work, we introduce a retrieval-aware multimodal tuning framework which can significantly improve the multimodal long document understanding ability of models.

## 6.2 Document Understanding Datasets

Given the practical and business applications of document understanding, researchers have devoted significant effort to this area by introducing new datasets and methods. SearchQA (Dunn et al., 2017), NarrativeQA (Kočiskỳ et al., 2018), QuALITY (Zhu et al., 2020) are reading comprehension datasets over purely textual data with an average length ranging from 1850 to 60k tokens. FinQA (Chen et al., 2021), DocFinQA (Reddy et al., 2024) are introduced in the financial domain. MarkQA (Huang et al., 2023) tackes QA over knowledge bases with numerical reasoning. DocVQA(Mathew et al., 2020) presents a visual question answering dataset on document images. VisualWebBench (Liu et al., 2024) is a multimodal benchmark over single-page documents focusing on various QA-style tasks. MMLongBench (Ma et al., 2024) is a multimodal document understanding dataset with an average of 47.5 pages and 21k textual tokens. Methods such as PDFTriage (Saad-Falcon et al., 2023) enables models to retrieve the context from long and structured documents. TAT-LLM (Zhu et al., 2024) addresses QA over a hybrid of tabular and textual data. ChartQA (Masry et al., 2022) is a benchmark of extractive questions-answering task over a chart image, while Chocolate (Huang et al., 2024b) annotates the types of factual errors in machine-generated chart captions.

Concurrent to our work, DocBench (Zou et al., 2024) is similar to MMLongBench as it also focuses on questions with short or extractive answers. In contrast, our benchmark mainly considers longer, open-ended answers which require more thorough understanding of the document. Furthermore, the authors have found that multimodal models such as GPT-4o perform worse than text-only GPT-4, which indicates the benchmark may be less suitable for multimodal evaluation. On the other hand, our results show that multimodal content is critical for our benchmark, as text-only inputs leads to significant performance degradation. Compared to the datasets above, our benchmark contains lengthier documents with hundreds of pages and focusing on open-ended questions which require in-depth solutions. We further propose an automated and reliable evaluation framework to assess the correctness of model answers, which demonstrates very high agreement with human preferences.

## 6.3 Retrieval-Augmented Generation

While recent multimodal models have shown impressive capability in many tasks, applying them directly to long document understanding tasks may face several challenges to the diverse multimodal content. Additionally, processing entire documents with large models is often impractical, as the text alone may contain millions of tokens, leading to substantial computational costs. Therefore, researchers have designed various retrieval augmented generation (Lewis et al., 2020; Chen et al., 2022) methods to address the issues. In this work, we have investigated multiple retrieval methods optimized for document page retrieval, including JINA-CLIP (Xiao et al., 2024) BM25 (Robertson & Zaragoza, 2009), BGE-M3 (Chen et al., 2024a) and ColPali (Faysse et al., 2024). However, despite using retrieval, existing multimodal models are constrained by their multimodal biases (Chen et al., 2024b) and susceptibility to irrelevant content that is inherent in the retrieved context (Shi et al., 2023). Thus, we further proposed a retrieval-aware tuning framework to enhance the performance of models when leveraging retrieval for multimodal long documents.

## 7 Conclusion

In this work, we introduce M-LongDoc, a benchmark dataset consisting of 851 samples and an automated framework to evaluate the performance of large multimodal models on document question answering tasks. This benchmark is specifically designed for long and diverse document formats containing text, figures, and tables, aligning with the demands of real-world applications. Unlike existing benchmarks, M-LongDoc features more recent and lengthy documents, often hundreds of pages long, and requires open-ended solutions rather than just extractive answers.

We also propose a retrieval-aware tuning approach designed for the efficient and effective processing of multimodal long documents. To our knowledge, this is the first training framework and model to directly address the retrieval setting for such documents. Experimental results demonstrate that our method achieves a relative improvement of 4.6% in the correctness of model responses compared to baseline open-source models. This improvement showcases the effectiveness of our approach in handling lengthy and complex multimodal documents, potentially aiding humans in various business and practical applications that require understanding and answering questions over such documents.

ETHICS STATEMENT

We will release the benchmark and training dataset publicly to facilitate further research in this area. To observe copyright rules, we do not release the documents directly, but instead the links to download each document. All annotators in this work were volunteers. While we focus on how models may answer questions based on multimodal documents, it is still possible for them to hallucinate information that is false or not verifiable.

REPRODUCIBILITY STATEMENT

In this work, we have included the details of our training framework and hyperparameters in Section 3 and 4. As discussed above, our benchmark dataset and questions will be released under a public licence. For reproducibility, our code will be found at `https://anonymous.4open.science/r/private-multimodal-documents-B2CF/`.

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

# A APPENDIX

## A.1 DATA VERIFICATION

To verify each question in our data construction process, we use the following guide to prompt the question generator models for automated verification. Similarly, we use the same guide for human annotation in the human verification stage.

Based on the document content and question, answer yes or no only to the following questions:

1. Does the content contain any {category}? 2. Does the question require information from the {category}? 3. Is the question clear and answerable based on the {category}? 4. Is the question of reasonable difficulty and answer cannot be simply copied?

Where {category} refers to table or figure or text, which is denoted with the question.

Note: If questions require general knowledge or commonsense in addition to the content, it is still acceptable. In the document PDF file, each question is shown with the ID corresponding to excel sheet, and the document page as image In the excel sheet, indicate "yes" or "no" for each check.

## A.2 RETRIEVAL METHODS

To support our retrieval-based document question answering setting, we currently include four state-of-the art methods to retrieve relevant pages based on each question. They include text-based sparse methods such as BM25 (Robertson & Zaragoza, 2009) embedding-based methods such as BGE-M3 (Chen et al., 2024a), multimodal piece-wise embedding methods such as JINA-CLIP (Xiao et al., 2024), and multimodal page-wise embedding methods such as ColPali (Faysse et al., 2024). Note that piece-wise embedding methods separate encode each piece of text, table image, or figure image, whereas page-wise methods can encode the entire page content as a single image. Thus, we rank each page in the document based on the similarity score or relevance score of that page with respect to the given question. As each page may have multiple pieces of content, we consider the highest score of all pieces in a page to be the page-wise relevance score. To compare the effectiveness of each method, we implement a standardized MRR score which refers to the mean reciprocal rank of the gold evidence page for each question. Based on the results in Table 5, we find that ColPali which encodes each page as single image shows the best performance. Thus, we select ColPali as the preferred retrieval method in our main experiments.

| Retriever | Text | Figure | Table | All |
|---|---|---|---|---|
| BM25 | 56.2 | 31.2 | 42.0 | 43.1 |
| CLIP | 57.1 | 37.9 | 50.4 | 48.5 |
| BGE-M3 | 66.4 | 36.4 | 53.6 | 52.1 |
| ColPali | 68.7 | 67.5 | 65.9 | 67.4 |

Table 5: Retriever performance comparison.

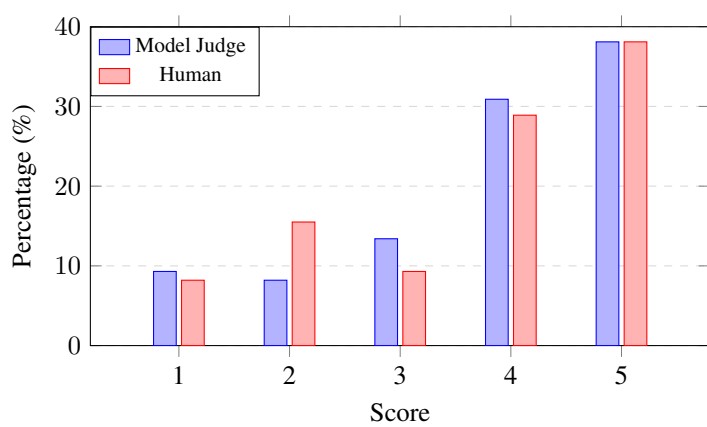

Figure 7: Distribution of evaluation scores by model judges and human annotators.

### A.3 EVALUATION GUIDE

To evaluate each model answer, we use the following scoring guide. Similarly, we use the same guide for human annotation in our analysis.

You will be given one response to a question based on a multimodal document containing texts, figures, or tables. Your task is to rate the response on correctness using a 1-5 scale. Please read and understand these instructions carefully, and keep them open for reference while reviewing.

Correctness (1-5) refers to how accurately, comprehensively, and appropriately the response addresses the question based on the information in the document.

5 - Fully Correct: Completely accurate, comprehensive, fully integrates relevant information from all parts of the document, and provides a coherent answer.

4 - Mostly Correct: Largely accurate with only minor errors or omissions, addresses most main points, and integrates information well.

3 - Partially Correct: Contains a mix of accurate and inaccurate information, addresses some key points but misses others, and partially integrates information.

2 - Mostly Incorrect: Has multiple inaccuracies, addresses only a small portion correctly, and shows minimal integration of information.

1 - Completely Incorrect: Contains significant errors, is irrelevant, or fails to address the question based on the document.

Evaluation Steps: 1. Thoroughly review the multimodal document and question. 2. Carefully read the response, comparing it to the document information. 3. Assess the response's accuracy, comprehensiveness, and relevance. 4. Assign a correctness score from 1 to 5 based on the criteria.

Question: question Response: answer

Evaluation Form (score only without explanation) Correctness:

### A.3.1 COMPARISON OF MODEL JUDGE AND HUMAN SCORES

### A.4 MORE EXAMPLES

### A.4.1 EXAMPLE OF M-LONGDOC

To investigate the diversity and in-depth nature of questions in our dataset, we manually categorize 100 random examples based on five common types, as shown in Table 6. Furthermore, Table 8 illustrates an example of a challenging question in our M-LongDoc benchmark. This question tests the ability of the model to identify and analyze trends across different charts and draw meaningful

| Category | Description | Proportion | Example Question |
|---|---|---|---|
| Analytical Reasoning and Pattern Recognition | Questions about trends, comparisons, and implications (e.g., engagement trends, performance trends) | 49% | What is the total amount of financial liabilities at amortized cost for the year 2023, and how does it compare to the total amount for 2022? Consider the implications of any changes in these liabilities on the company's financial strategy. |
| Technical Analysis | Questions about specific technical details (e.g., UEFI BIOS, shutter speeds, X-sync speeds) and applications of technical concepts. | 37% | What potential issue could arise if you fail to follow the instruction to tighten the screws twice when installing the top cover, and why might this step be particularly important for a laptop? |
| Commonsense and Domain Knowledge | Questions requiring general knowledge or background knowledge in fields such as finance, cybersecurity, photography. | 46% | What are the key differences and potential advantages of using white-box analysis over machine learning for modeling the performance of configurable systems, as discussed by Velez et al. (2021)? |
| Visual Interpretation | Questions based on interpreting icons, diagrams, or charts. | 60% | Explain the functionalities of the different sections (a, b, c, d) in the LaserFactory design toolbar and discuss how each section contributes to the overall design and fabrication process. |
| Mathematical Reasoning | Questions involving mathematical concepts or calculation from data. | 17% | Calculate the percentage change in diluted net income per share attributable to common stockholders from fiscal year 2023 to fiscal year 2024. What factors likely contributed to this change? |

Table 6: Categorization of question types with descriptions and examples.

| Dataset | Avg. Question Length | Avg. Answer Length |
|---|---|---|
| DocVQA | 8.5 | 2.4 |
| MMLongBench-Doc | 16.4 | 2.6 |
| M-LongDoc (Ours) | 31.6 | 180.3 |

Table 7: Comparison of average question length and answer length in tokens for different datasets.

comparisons. To further compare with existing datasets, we measure the average question lengths and answer lengths as shown in Table 7.

### A.4.2 CASE STUDY OF RETRIEVAL-AWARE TUNING

Table 9 displays a sample question in M-LongDoc and the answers generated by Qwen2-VL and Qwen2-VL w/ Retrieval-aware Tuning. The answer generated by Qwen2-VL states that the Cosine method consistently shows the highest latent cosine similarity across all datasets, which is incorrect. In fact, the zero-shot stitching experiment does not involve the Cosine method. It appears that Qwen2-VL may have been misled by the keyword "cosine" appearing elsewhere in the retrieved context. In contrast, the answer generated by Qwen2-VL w/ Retrieval-aware Tuning correctly identifies that the affine method consistently obtains the highest latent cosine similarity (lcos) across all datasets. This demonstrates the effectiveness of our Retrieval-aware Tuning method in improving the model's capability to comprehend retrieved context.

**Question:**

How does the relationship between reference length percentile and the percentage of empty modes differ from the relationship between reference sentence length percentile and the probability of empty context? Explain the key differences in the trends shown by these two graphs.

**Relevant page (truncated):**

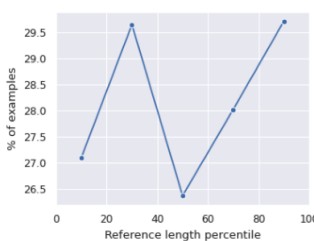
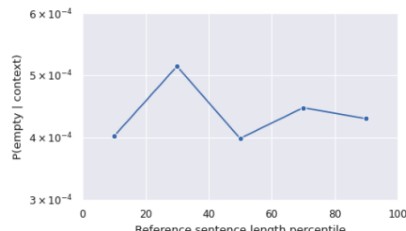

(a) Percent of stories that have the empty sequence as their modal continuation.

(b) Geometric mean of the model's probability of the empty sequence given the first four sentences of the story.

Figure 6.2: Finetuned GPT-2-345M predictions of empty outputs on the ROC Stories validation set (1571 Stories). Stories are grouped into 5 equally sized bins by reference continuation length.

Table 6.2: Modal continuations of several lengths for prefix: "Sarah always had a fascination with the night sky. Noticing her passion, Sarah's father bought her a new telescope. She was ecstatic. She went outside every night to diligently view the night sky." The reference continuation is "Sarah loved her new telescope."

| Length Constraint (tokens) | Log-probability | Text |
| --- | --- | --- |
| Global mode | -7.79 | `<|endoftext|>` |
| 5 | -9.14 | Sarah loved astronomy!`<|endoftext|>` |
| 6 | -7.97 | Sarah never looked back.`<|endoftext|>` |
| 7 | -8.59 | Sarah loved her new telescope.`<|endoftext|>` |
| 8 | -9.38 | Now, Sarah is an astronomer.`<|endoftext|>` |
| 9 | -8.68 | Sarah was happy with her new telescope.`<|endoftext|>` |
| 10 | -8.77 | Sarah was very happy with her new telescope.`<|endoftext|>` |
| 12 | -8.91 | Sarah was amazed by the beauty of the night sky.`<|endoftext|>` |

An interesting feature of these constrained modes is that the content can be correlated with the length in clear ways. Table 6.3 shows an example where the mode of length 5 is significantly different from all the other modes. It may be impossible to produce a 5 token output that has the right content, but the model "prefers" to output something grammatical, so we see different content. This is different from the short NMT modes, which were often truncated when the constraint was too short to express the content of the source sentence.

In order to show that these patterns aren't just cherry-picked, randomly sampled examples of modal outputs are shown in Table B.2. All 30 of the conditional modes are grammatical, relevant to the context, and don't show any evidence of degenerate behavior. This is further evidence that conditional MAP inference may be a promising direction of investigation.

Table 8: An example of a challenging question from M-LongDoc that requires the model to compare the trends of two charts in a document.

**Question:**
Which method consistently shows the highest latent cosine similarity (lcos) across all datasets (MNIST, F-MNIST, CIFAR-10, CIFAR-100) in the zero-shot stitching experiment, and what might this imply about the nature of transformations between latent spaces in autoencoders?

**Answer generated by Qwen2-VL:**
The method that consistently shows the highest latent cosine similarity (lcos) across all datasets (MNIST, F-MNIST, CIFAR-10, CIFAR-100) in the zero-shot stitching experiment is the "Cosine" method. This implies that the "Cosine" method is able to capture the most similar latent spaces across different architectures, modalities, and datasets. This suggests that the nature of transformations between latent spaces in autoencoders is not exclusively correlated by orthogonal transformations, but rather by a class of transformations larger than orthogonal transformations. This finding highlights the importance of understanding and modeling the particular class of transformations that control the correlation between NNs during image autoencoding.

**Answer generated by Qwen2-VL w/ Retrieval-aware Tuning:**
The method that consistently shows the highest latent cosine similarity (lcos) across all datasets (MNIST, F-MNIST, CIFAR-10, CIFAR-100) in the zero-shot stitching experiment is the "affine" method. This method achieves the highest lcos values for all datasets, with scores of 0.94 for MNIST, 0.94 for F-MNIST, 0.96 for CIFAR-10, and 0.96 for CIFAR-100. This consistency in high lcos values suggests that the affine transformation is particularly effective in aligning the latent spaces of different autoencoders. The high lcos indicates a strong alignment between the latent representations of the autoencoders, which implies that the affine transformation is capable of capturing the underlying structure and relationships between the latent spaces more accurately than other methods. The results imply that the nature of transformations between latent spaces in autoencoders is not exclusively controlled by orthogonal transformations. The affine transformation, which includes scaling, rotation, and translation, appears to be a more versatile and effective method for aligning these spaces. This finding suggests that the latent spaces of image autoencoders are related by a broader class of transformations, including those that are not strictly orthogonal, which is a significant insight into the nature of these transformations and their implications for model stitching and zero-shot learning.

**Relevant page (truncated):**

*5.3. Latent Communication via Translation*       39

TABLE 5.3: Zero-shot stitching for generation with various methods for estimating $\widehat{\mathcal{T}}$. The representation is normalized using Standard Scaling, and no additional normalization is applied to the stitched decoders. We report the latent cosine similarity (*lcos*) and MSE (*lmse*) between the target encoding and the translated one, but also the reconstruction MSE (*rmse*) between the input and the output. The absolute space dimension is 500, and we used 1000 anchors. Please refer to Table B.1 for results on decoders scale-invariant by design (with L2 normalization on the encodings).

| | MNIST | | | F-MNIST | | | CIFAR-10 | | | CIFAR-100 | | |
|---|---|---|---|---|---|---|---|---|---|---|---|---|
| | *lcos* | *lmse* | *rmse* | *lcos* | *lmse* | *rmse* | *lcos* | *lmse* | *rmse* | *lcos* | *lmse* | *rmse* |
| absolute | 0.09 | 0.27 | 0.14 | 0.17 | 0.23 | 0.23 | 0.30 | 0.29 | 0.34 | 0.34 | 0.53 | 0.40 |
| affine | 0.94 | 0.08 | 0.02 | 0.94 | 0.06 | 0.03 | 0.96 | 0.03 | 0.05 | 0.96 | 0.04 | 0.05 |
| linear | 0.92 | 0.09 | 0.02 | 0.93 | 0.07 | 0.04 | 0.94 | 0.03 | 0.05 | 0.94 | 0.04 | 0.06 |
| l-ortho | 0.79 | 0.14 | 0.02 | 0.78 | 0.12 | 0.05 | 0.85 | 0.05 | 0.06 | 0.84 | 0.07 | 0.07 |
| ortho | 0.90 | 0.10 | 0.02 | 0.90 | 0.08 | 0.04 | 0.94 | 0.03 | 0.06 | 0.93 | 0.04 | 0.06 |

in structure, differing only in the random seed used for weight initialization and data shuffling. To perform Zero-Shot Stitching, we first translate each data point from the latent space of the first encoder to the latent space of the second using 1000 parallel anchors. We then apply the second decoder to the translated data, without any additional training or fine-tuning.

**Result analysis.** This experiment analyzes the alignment of latent spaces in different training regimens of the same AE. The performance evaluation, as shown in Table 5.3, demonstrates that all methods affine, linear, l-ortho, and ortho yield satisfactory results. Moreover, qualitative results depicted in Figure 5.6 reveals minimal visual differences in the stitching outcomes across various datasets using different methods. Please refer to Figures B.4 and B.5 for other qualitative results. In fact, these

Table 9: Sample answers generated by Qwen2-VL and Qwen2-VL w/ Retrieval-aware Tuning, respectively.

