# OpenReview forum: "M-Longdoc: A Benchmark For Multimodal Super-Long Document Understanding And A Retrieval-Aware Tuning Framework"
_ICLR.cc/2025/Conference — Submitted to ICLR 2025_

### Official Review · Reviewer_t1TM · 2024-10-30

**Soundness:** 3
**Presentation:** 3
**Contribution:** 2
**Rating:** 5
**Confidence:** 4

**Summary:**

The paper introduces M-LongDoc. M-LongDoc is a novel benchmark dataset to evaluating multimodal long document (210 pages in average) understanding. M-LongDoc features 851 samples that challenge existing multimodal models / systems to answer open-ended questions requiring in-depth understanding of complex documents (including financial reports and academic papers).

Besides the benchmark datasets, the paper offers a retrieval-aware tuning framework to explore the solutions to solve the problem. The retrieval-augmented tuning approach improves model performance by guiding attention to relevant content while ignoring distractions.

**Strengths:**

1. M-LongDoc provides a new benchmark. It is different from existing multimodal document visual question answering benchmarks, such as MP-DocVQA and DUDE, which is much longer (210 pages in average), and they are all open-ended questions (not extractive QA or short answer QA).

2.  The paper proposes a retrieval-aware tuning to improve multimodal model performance on M-LongDoc benchmarks, a strategy that could benefit applications requiring nuanced document comprehension.

**Weaknesses:**

1. All questions in the benchmark are synthetically generated by multimodal LLMs, which may limit the benchmark's reflection of real-world scenarios. Human annotations are not involved in the benchmark creation process.

2. The dataset's scale is relatively modest (only 852 samples), potentially insufficient for capturing a wide range of perspectives and real-world scenarios.

3. Evaluation relies on proprietary LLMs, introducing potential variability due to different checkpoints or versions.

4. Some related works are missing [1], the differences are not discussed.

[1] DocBench: A Benchmark for Evaluating LLM-based Document Reading Systems

**Questions:**

1. How would direct text extraction and retrieval perform as an alternative to solve the problem? What are the performance on text-only questions? What are the performance on multimodal questions?

2.  Retrieval tuning appears to yield only marginal performance gains. What specific challenges in retrieval are contributing to this limited improvement, and how do they align with the unique requirements of M-LongDoc?

---

> ### Author Response · Authors · 2024-11-20
> **Response to weaknesses**
>
> > 1. All questions in the benchmark are synthetically generated by multimodal LLMs, which may limit the benchmark's reflection of real-world scenarios. Human annotations are not involved in the benchmark creation process.
>
> While it is true that the questions in M-LongDoc were model-generated, we want to emphasize that all questions underwent rigorous human verification and quality control before inclusion in the final benchmark. Our team of expert annotators carefully reviewed each generated question to ensure relevance, appropriate content, and comprehensiveness. (lines 194-203). Furthermore, we believe that the generated questions are diverse and reflect real-world scenarios. Based on the example below from Figure 3, the question about oven vents is something a real person might ask when using or purchasing an appliance, related to everyday life and home cooking.
>
> | Dataset | Example Question | Example Answer |
> |---------|------------------|-----------------|
> | DocVQA | What is the underlined heading just above the table? | Indications for implantation |
> | MMLongBench | What is the number of red logos in page 10? | 0 |
> | M-LongDoc (Ours) | Where are the oven vents located on this range model, and why is their positioning important for proper oven function? | The oven vents are located at the top front of the oven, with one vent on the upper front and another on the lower front. Their positioning is important for proper oven function because they release hot air and moisture from the oven during cooking and cleaning. Blocking or covering the vents can cause poor air circulation, affecting cooking and cleaning results. The vents also help to maintain a consistent temperature in the oven by releasing excess heat and preventing the oven from overheating. |
>
> > 2. The dataset's scale is relatively modest (only 852 samples), potentially insufficient for capturing a wide range of perspectives and real-world scenarios.
>
> While the M-LongDoc dataset contains 851 samples, which may seem modest, this size is comparable to other similar multimodal document understanding benchmarks, while requiring more in-depth answers. For example, MMLongBench-Doc contains 1051 samples. The controlled data size allows for careful human verification of each sample, ensuring high quality. Additionally, creating a larger dataset for long, complex documents with open-ended questions is extremely resource-intensive. We believe this approach balances scale and quality, providing sufficient samples to evaluate model capabilities while maintaining rigorous standards. The diverse document domains and question categories based on different multimodal elements provide a rich testbed, as shown in Table 1.
>
> > 3. Evaluation relies on proprietary LLMs, introducing potential variability due to different checkpoints or versions.
>
> While there may be some variability in different model versions, we have specified the fixed versions of gpt-4o-2024-05-13, claude-3-5-sonnet-20240620, and gemini-1.5-pro-002  in our paper for reproducibility (lines 407-420). The reason for selecting proprietary models is that they demonstrate leading performance (lines 241-243) compared to open-source models and are widely used in model-based evaluation, such as in “G-Eval: NLG Evaluation using GPT-4 with Better Human Alignment”.
>
> > 4. Some related works are missing [1], the differences are not discussed. [1] DocBench: A Benchmark for Evaluating LLM-based Document Reading Systems
>
> Thank for you mentioning this related work. While it is relevant, we also note that it was released on arXiv in July 2024, which is considered concurrent work according to ICLR guidelines. Regarding DocBench, it is similar to the existing MMLongBench-Doc as it also focuses on questions with short or extractive answers. In contrast, our benchmark mainly considers longer, open-ended answers which require more thorough understanding of the document. Furthermore, the authors of DocBench have found that multimodal models such as GPT-4o perform worse than text-only GPT-4, which indicates the benchmark may be less suitable for multimodal evaluation. On the other hand, our results from Table 4 below show that multimodal content is critical for our benchmark, as text-only inputs leads to significant performance degradation. We will include the discussion of this concurrent related work in the revised version.
>
> | Model | Text | Figure | Table |
> |-------|------|--------|-------|
> | Qwen2-VL | 4.08 | 3.83 | 3.62 |
> | w/ Text-only Inputs | 4.22 | 3.37 | 3.38 |

---

> ### Author Response · Authors · 2024-11-20
> **Response to questions**
>
> > 1. How would direct text extraction and retrieval perform as an alternative to solve the problem? What are the performance on text-only questions? What are the performance on multimodal questions?
>
> Thank you for raising this question. We assume that direct text extraction refers to using only the extracted text of each PDF document to generate the answer, without considering the visual contents of figures or tables. Regarding direct text extraction and retrieval, we separately consider text-only model inputs and text-only retrieval to avoid confounding analysis. Firstly, the results in Table 4, also attached below, show the effect of removing figure and table images from the model inputs, leaving only the extracted document text as input. Specifically, in this text-only setting, performance increases slightly for text-based questions, suggesting that for text-only questions, the visual content in the document may mislead the model in rare cases (lines 454-463). On the other hand, we found significantly lower scores in multimodal questions (figure-based and table-based), indicating that  the direct text extraction alone cannot support multimodal questions well. Please let us know if you intended a different meaning of the question.
>
> | Model | Text | Figure | Table |
> |-------|------|--------|-------|
> | Qwen2-VL | 4.08 | 3.83 | 3.62 |
> | w/ Text-only Inputs | 4.22 | 3.37 | 3.38 |
>
> Regarding text-only retrieval, we compare the MRR score of different retrievers in Table 5 below. In this comparison, BM25 and BGE-M3 only consider document text for retrieval, while CLIP and ColPali can consider both text and images. We find that Colpali as a multimodal retriever significantly outperforms BGE-M3 as a text-only retriever, especially in figure-based and table-based questions. However, we note that text-only retrievers and multimodal retrievers tend to have different training data and model architectures specific to each setting. Thus, it may not be fair to make general conclusions regarding text-only compared to multimodal retrieval.
>
> | Retriever | Text | Figure | Table |
> |-----------|------|--------|-------|
> | BM25      | 56.2 | 31.2   | 42.0  |
> | CLIP      | 57.1 | 37.9   | 50.4  |
> | BGE-M3    | 66.4 | 36.4   | 53.6  |
> | ColPali   | 68.7 | 67.5   | 65.9  |
>
> > 2. Retrieval tuning appears to yield only marginal performance gains. What specific challenges in retrieval are contributing to this limited improvement, and how do they align with the unique requirements of M-LongDoc?
>
> Thank you for raising this question. To analyze the challenges in retrieval, we evaluated different retrieval settings in the table below. Based on the results, we believe the key challenge is the model’s ability to leverage relevant information from the retrieved pages (lines 359-362). To consider the ideal case of retrieval, we evaluate our finetuned model and ensure that the retrieved pages contain the gold evidence page (top-k=5 including gold page). In this case, we observe a slight improvement of +0.06 compared to standard retrieval with our finetuned model (top-k=5 retrieved pages). On the other hand, to consider the upper bound of model performance, we consider the oracle setting where the model is only provided the gold evidence page (no retrieval, gold page only). This results in a noticeable improvement of +0.20 compared to standard retrieval with our finetuned model.
>
> Thus, while our tuning approach does show a significant improvement of +0.18 (4.6%) compared to the base model, there is also a significant gap to reach the upper bound performance with respect to retrieval. This suggests that distinguishing between relevant and irrelevant content in the retrieved pages in a key bottleneck. This challenge is core to the requirements of M-LongDoc, as our benchmark presents very long multimodal documents with hundreds of pages, which are not practical to fully process with large models. Thus, the ability to leverage retrieval and generate the correct answer based on the relevant pages is crucial to an effective document understanding system. We will include this analysis in the revised version, and aim to further study this challenge in future work.
>
> | Model | Retrieval Setting | All |
> |------|-------------------|-----|
> | Qwen2-VL | Top-k=5 retrieved pages | 3.84 |
> | Qwen2-VL w/ Retrieval Tuning | Top-k=5 retrieved pages | 4.02 |
> | Qwen2-VL w/ Retrieval Tuning | Top-k=5 including gold page | 4.08 |
> | Qwen2-VL w/ Retrieval Tuning | No retrieval, gold page only | 4.22 |

---

> > ### Comment · Reviewer_t1TM · 2024-11-25
> >
> > Thanks for your response, I decided to keep my original score, due to
> >
> > 1. I am still concerned about data size and synthetic questions, which may introduce bias in using this benchmark to evaluate LLMs.
> >
> > 2. Only using text also achieves similar performance with multimodal information, which is hard to justify the multimodal usage in answering the question.
> >
> > 3. Lack for comparison with related work (not reflected in the revised version)

---

> > > ### Author Response · Authors · 2024-11-26
> > >
> > > > 1. I am still concerned about data size and synthetic questions, which may introduce bias in using this benchmark to evaluate LLMs.
> > >
> > > Regarding data size and synthetic questions: We understand your concern about potential bias. However, we believe the careful human verification process and diverse questions help mitigate this risk. For example, we manually categorized 100 questions in the table below, showing that they cover realistic aspects involving analytical reasoning, technical analysis, commonsense and domain knowledge, visual interpretation and mathematical reasoning, where many questions can involve multiple aspects. As shown in [Figure 1](https://imgur.com/a/878wjtQ), the dataset, while not extremely large, provide diverse coverage across different domains and topics. While we observe that some well-establised benchmarks such as MT-Bench [1] and TruthfulQA [2] also contain fewer questions, we aim to expand the dataset and incorporate human-generated questions in future work. We have also included this analysis in the revised version (lines 860-861), thank you.
> > >
> > > [1] Zheng, Lianmin, et al. "Judging llm-as-a-judge with mt-bench and chatbot arena." Advances in Neural Information Processing Systems 36 (2023): 46595-46623.
> > >
> > > [2] Stephanie Lin, Jacob Hilton, and Owain Evans. 2022. TruthfulQA: Measuring How Models Mimic Human Falsehoods. In Proceedings of the 60th Annual Meeting of the Association for Computational Linguistics (Volume 1: Long Papers), pages 3214–3252, Dublin, Ireland. Association for Computational Linguistics.
> > >
> > > | Category | Description | Proportion | Example Question |
> > > |----------|-------------|------------|-------------------|
> > > | Analytical Reasoning and Pattern Recognition | Questions about trends, comparisons, and implications (e.g., engagement trends, performance trends) | 49% | What is the total amount of financial liabilities at amortized cost for the year 2023, and how does it compare to the total amount for 2022? Consider the implications of any changes in these liabilities on the company's financial strategy. |
> > > | Technical Analysis | Questions about specific technical details (e.g., UEFI BIOS, shutter speeds, X-sync speeds) and applications of technical concepts. | 37% | What potential issue could arise if you fail to follow the instruction to tighten the screws twice when installing the top cover, and why might this step be particularly important for a laptop? |
> > > | Commonsense and Domain Knowledge | Questions requiring general knowledge or background knowledge in fields such as finance, cybersecurity, photography. | 46% | What are the key differences and potential advantages of using white-box analysis over machine learning for modeling the performance of configurable systems, as discussed by Velez et al. (2021)? |
> > > | Visual Interpretation | Questions based on interpreting icons, diagrams, or charts. | 60% | Explain the functionalities of the different sections (a, b, c, d) in the LaserFactory design toolbar and discuss how each section contributes to the overall design and fabrication process. |
> > > | Mathematical Reasoning | Questions involving mathematical concepts or calculation from data. | 17% | Calculate the percentage change in diluted net income per share attributable to common stockholders from fiscal year 2023 to fiscal year 2024. What factors likely contributed to this change? |
> > >
> > >
> > > > 2. Only using text also achieves similar performance with multimodal information, which is hard to justify the multimodal usage in answering the question.
> > >
> > > Text-only vs multimodal performance: We apologize if this wasn't clear in our previous response. As shown in Table 4, we believe there is a significant performance drop for figure-based (-12.0%) and table-based (-6.6%) questions when using text-only inputs. This strongly indicates the importance of multimodal information for these question types. On the other hand, the text-only model may be able to answer multimodal questions to a limited extent, as the text may contain partial information about the tables and figures (lines 458-460). We have revised the explanation in section 5.2 to clarify this analysis (457-460).
> > >
> > > > 3. Lack for comparison with related work (not reflected in the revised version)
> > >
> > > Comparison with related work: We appreciate you bringing DocBench to our attention. As mentioned, it was concurrent work, but we agree it's valuable to discuss. We have revised the paragraph in the related work section to compare our approach to DocBench (lines 501-507), highlighting key differences like our focus on open-ended questions requiring deeper understanding, against their shorter or extractive answers. We have also noted how our benchmark demonstrates benefits of multimodal inputs, unlike their previous findings.
> > >
> > > We hope these clarifications address your concerns. We thank the reviewer for their feedback and are committed to improving the paper and benchmark. Please let us know if you have any further questions or suggestions.

---

> > > > ### Comment · Reviewer_t1TM · 2024-11-26
> > > >
> > > > Thanks authors for the further clarification, I decided to maintain my original score.

---

> > > > > ### Author Response · Authors · 2024-11-27
> > > > >
> > > > > > Thanks authors for the further clarification, I decided to maintain my original score.
> > > > >
> > > > > We thank the reviewer for the response, and respectfully ask if you have any specific concerns remaining, such as regarding our data or findings? We would be happy to continue the constructive discussion, addressing your concerns to the best of our ability. To provide a clearer picture of our dataset, we have also made an anonymous link of 100 question samples for your review, thank you.
> > > > >
> > > > > https://docs.google.com/spreadsheets/d/e/2PACX-1vR3jH8LqkERk9oPM8T_s2xVGJ_tQKcP5n7aRlVyu1eyjOTMRUQGhEZ29kJx6HgDSkTt85QhgHms_QUg/pubhtml

---

> > > > > ### Author Response · Authors · 2024-12-01
> > > > >
> > > > > Dear reviewer,
> > > > >
> > > > > Thank you for your previous responses, and we have provided further analysis and revisions based on your valuable suggestions above. However, it is close approaching the end of the discussion period. Could we kindly enquire if our responses and adjustments have adequately resolved your concerns? We are more than happy to answer any further queries or concerns you may have. Thank you once again.

---

> > > > > > ### Comment · Reviewer_t1TM · 2024-12-01
> > > > > >
> > > > > > I dont have more concerns

---

> > > > > > > ### Author Response · Authors · 2024-12-01
> > > > > > >
> > > > > > > Dear Reviewer,
> > > > > > >
> > > > > > > Thank you for your time and attention throughout this process. We greatly appreciate your thorough feedback and the opportunity to address your concerns. We were particularly encouraged by your recognition of the unique contributions, including:
> > > > > > > 1. Introducing a new benchmark for multimodal long document understanding, distinct from existing datasets in its length and open-ended question format.
> > > > > > > 2. Proposing a retrieval-aware tuning approach for multimodal long document question answering, which can benefit applications requiring nuanced document comprehension.
> > > > > > >
> > > > > > > Given that you've indicated you have no further concerns, we were wondering if you might be willing to reconsider your score in light of the revisions and clarifications we've provided. We believe that addressing your initial concerns, combined with the strengths you've identified, may warrant a more favorable evaluation.
> > > > > > >
> > > > > > > If you feel our responses have adequately addressed the initial issues and the paper's contributions are valuable to the field, an updated score would be very helpful in accurately reflecting the current state of our work. We appreciate your consideration and look forward to your response.

---

> > > > > > > ### Author Response · Authors · 2024-12-02
> > > > > > >
> > > > > > > Dear Reviewer,
> > > > > > >
> > > > > > > As we approach the end of the discussion period, we wanted to follow up on our previous response. We appreciate your earlier feedback indicating you had no further concerns. Given that today is the final day for discussion, we kindly ask if you could provide any last thoughts or, if you feel it's warranted based on our revisions and clarifications, consider updating your rating score.
> > > > > > >
> > > > > > > Your final input would be invaluable in ensuring a comprehensive review process. Thank you again for your time and expertise throughout this discussion.

---

> > > > > > > ### Author Response · Authors · 2024-12-03
> > > > > > >
> > > > > > > Dear Reviewer,
> > > > > > >
> > > > > > > We sincerely apologize for reaching out again. As we approach the end of the discussion period, we wanted to follow up on our previous response. We noticed you indicated that your concerns have been resolved, and we were hoping to request if you might consider increasing the score kindly.
> > > > > > >
> > > > > > > --
> > > > > > >
> > > > > > > Authors

---

> > > > > > > ### Author Response · Authors · 2024-12-03
> > > > > > >
> > > > > > > Dear reviewer, as this is the last day for reviewer responses, and you indicated that your concerns have been resolved, we kindly ask if you could provide any last thoughts or, if you feel it's warranted based on our revisions and clarifications, consider updating your rating score. Thank you again for your time and expertise throughout this discussion.

---

### Official Review · Reviewer_Jfm1 · 2024-11-03

**Soundness:** 3
**Presentation:** 3
**Contribution:** 3
**Rating:** 8
**Confidence:** 4

**Summary:**

The paper presents M-LongDoc, a new benchmark for evaluating the ability of large multimodal models to understand and answer open-ended questions over lengthy and diverse documents containing text, figures, and tables. M-LongDoc comprises 851 samples across academic, financial, and product domains, featuring documents significantly longer and more structurally complex than those in existing benchmarks. The authors also proposes a novel retrieval-aware tuning approach that specifically trains models to handle potentially irrelevant retrieved content, leading to a 4.6% relative improvement in answer correctness compared to baseline open-source models. Lastly, the authors contribute a large-scale training corpus of 10,070 samples and an automated evaluation framework based on a committee of multimodal judges to assess the correctness of open-ended solutions.

**Strengths:**

Useful new eval dataset, training dataset, eval framework and interesting model for multimodal RAG-QA on long docs.

**Weaknesses:**

This paper doesn't really have any major weaknesses. In particular, the paper presents its contribution as being primarily a dataset paper, so there's understandably not much novelty with the models.

**Questions:**

All clear to me

---

> ### Author Response · Authors · 2024-11-22
>
> We thank the reviewer for their positive review of our paper on M-LongDoc. Your feedback on the strengths of our work, particularly noting the usefulness of the new benchmark and training dataset and model for multimodal RAG-QA on long documents, is very encouraging.
>
> We are pleased that you found the paper to be sound, well-presented, and a good contribution to the field. Your observation that the paper doesn't have any major weaknesses aligns with our efforts to create a valuable and robust contribution to multimodal document understanding. Your feedback reinforces our belief in the importance of developing resources and methods for handling long, complex multimodal documents. We see this work as a stepping stone for future research in this area, potentially inspiring new approaches to multimodal document understanding and retrieval-augmented generation.

---

### Official Review · Reviewer_rjZp · 2024-11-04

**Soundness:** 3
**Presentation:** 3
**Contribution:** 2
**Rating:** 5
**Confidence:** 4

**Summary:**

This paper introduces a dataset for long document understanding challenges with an automatic evaluation approach. The authors also propose a retrieval-aware tuning framework to mitigate the limitations of current LLMs dealing with multimodal long documents.

**Strengths:**

- Research problem is challenging and necessary for this direction and various domain applications.
- A new dataset is proposed for multimodal long document understanding.
- A LLM-based auto-evaluating framework is introduced.

**Weaknesses:**

- The scope of this paper limited the question only focusing on specific pages ignoring the more natural cases of answers distributed spanning pages. And the author mentioned the in-depth but there is no supporting analysis and results to show why the dataset shows more in-depth.
- The dataset generation workflow uses off-the-shelf tools and models to extract the document structure which should be verified as the accumulated errors may occur when moving to the automatic QA generation stage.
- More dataset analyses are expected including question length and focusing topics. Simple statistics can not show more insight of your datasets.
- The proposed evaluation metrics may need more detailed analysis to show robustness. Current average weighting looks too simple ignoring the difference between specific models dealing with specific types of questions. Some penalty or reward terms may need to be considered.
- As the metrics are unexplored the results may not be comprehensive and reliable. Lack of quantitative analysis show different domain, question types performance.

**Questions:**

- Does your dataset consider the spanning-page answer setting?
- Is there a dataset structure parsing quality checking procedure?
- Is there any analysis or comparison before and after human checking automatically generated QA pairs?
- Why does the number of document pages per document look wired, especially for academic papers? It might be too long for an academic paper.

---

> ### Author Response · Authors · 2024-11-20
> **Response to weaknesses (paper scope, data generation, data analysis)**
>
> > 1a. The scope of this paper limited the question only focusing on specific pages ignoring the more natural cases of answers distributed spanning pages.
>
> Thank you for raising this point. We limited questions to single-page evidence for a more focused evaluation, requiring more comprehensive understanding of individual pages' contents before tackling multi-page reasoning. This approach enables the creation of a higher-quality dataset that is more straightforward to generate and evaluate at scale. While we also believe multi-page questions are an important future direction, the current setting remains challenging for open-source models and also reflects many real-world use cases. Overall, we believe the current setting addresses an important gap in multimodal long document understanding while laying the groundwork for more complex settings in future iterations.
>
> > 1b. And the author mentioned the in-depth but there is no supporting analysis and results to show why the dataset shows more in-depth.
>
> Based on the examples below which are included in our Figure 3, we show that questions in M-LongDoc are more complex than those from other benchmarks, as they requires a longer explanation or analysis rather than an extraction of a short text span or value (lines 125-130). Thus, we believe our benchmark is more in-depth as it requires more comprehensive and thorough understanding of each page in the document contents. This is further supported by the comparison on question lengths and answer lengths in the table below, showing that our questions and answers contain significantly more tokens on average. Thank you for raising this point, we will include this additional analysis in the revised version.
> | Dataset | Example Question | Example Answer |
> |---------|------------------|-----------------|
> | DocVQA | What is the underlined heading just above the table? | Indications for implantation |
> | MMLongBench | What is the number of red logos in page 10? | 0 |
> | M-LongDoc (Ours) | Where are the oven vents located on this range model, and why is their positioning important for proper oven function? | The oven vents are located at the top front of the oven, with one vent on the upper front and another on the lower front. Their positioning is important for proper oven function because they release hot air and moisture from the oven during cooking and cleaning. Blocking or covering the vents can cause poor air circulation, affecting cooking and cleaning results. The vents also help to maintain a consistent temperature in the oven by releasing excess heat and preventing the oven from overheating. |
>
> | Dataset | Avg. Question Length | Avg. Answer Length |
> |-------|------|--------|
> | DocVQA | 8.5 | 2.4 |
> | MMLongBench-Doc | 16.4 | 2.6 |
> | M-LongDoc (Ours) | 31.6 | 180.3 |
>
> > 2. The dataset generation workflow uses off-the-shelf tools and models to extract the document structure which should be verified as the accumulated errors may occur when moving to the automatic QA generation stage.
>
> While there may be some data extraction errors from off-the-shelf tools, we believe that our automated and human verification as discussed in Section 2.2 can address this concern. As each question addresses the specific text, table, or figure content in a document page, our verification also considers whether the specified data category is present in the extracted content, and whether the question can be answered based on the extracted content. Thus, each question is first automatically checked and then human verified to avoid errors from the data extraction tools. For reference, the full verification checklist is included in Appendix A.1 (lines 769 to 780).
> > 3. More dataset analyses are expected including question length and focusing topics. Simple statistics can not show more insight of your datasets.
>
> Based on the dataset, we find that the questions on average contain 31.6 tokens, which is significantly longer than other datasets, as shown in the table below. In addition to detailed statistics of the dataset, as shown in Figure 2, we illustrate the data topic distribution in [Figure 1](https://imgur.com/a/878wjtQ). The topics are obtained from the metadata of the documents, such as the paper category in the academic domain. As shown in Figure 1, the data domains span diverse topics such as machine learning, healthcare, and even laptop devices.
> | Dataset | Avg. Question Length | Avg. Answer Length |
> |-------|------|--------|
> | DocVQA | 8.5 | 2.4 |
> | MMLongBench-Doc | 16.4 | 2.6 |
> | M-LongDoc (Ours) | 31.6 | 180.3 |

---

> > ### Comment · Reviewer_rjZp · 2024-12-02
> >
> > Thanks for the detailed replies from the authors. There are many similar dataset papers extending the single-page document VQA scenarios to multiple pages. The contribution of this paper is not obvious. The answer is still located on a specific page ignoring the logical and semantic connection for long document understanding, especially for table and figure-related questions. This is the main concern of this paper from my side. Author concerns are addressed by authors well. I'm happy to increase my mark.

---

> > > ### Author Response · Authors · 2024-12-02
> > >
> > > Dear Reviewer,
> > >
> > > Thank you for your thoughtful feedback and for considering our responses. We greatly appreciate your willingness to increase your mark based on our clarifications. We understand your concern about the questions still being localized to specific pages rather than spanning multiple pages. While we agree that multi-page reasoning is an important direction for long document understanding, we believe our current data setting still offers several key contributions compared to previous work:
> > > 1. It establishes an important baseline for in-depth multimodal understanding of each page before tackling cross-page connections. This allows us to isolate and evaluate text-based, figure-based, and table-based questions in diverse domains.
> > > 2. A key focus of our multimodal document benchmark is open-ended questions requiring longer answers, with aspects such as analytical reasoning, technical analysis, commonsense knowledge or math reasoning, as discussed in the previous responses.
> > > 3. It enables creation of a higher-quality dataset that is more straightforward to generate and evaluate at scale, while still being challenging for current models.
> > >
> > > Additionally, our work offers:
> > > - An automated and reliable evaluation framework to facilitate research on open-ended multimodal document question answering.
> > > - Detailed analyses of leading models, revealing their multimodal bias and practical challenges when leveraging retrieval methods.
> > > - A retrieval-aware tuning approach to improve model performance by up to 4.6%, a strategy that can benefit many applications for long multimodal documents.
> > >
> > > We believe these provide meaningful contributions to advance multimodal long document understanding research. That said, we appreciate your perspective on the importance of cross-page connections. In future work, we plan to extend our dataset to include multi-page reasoning questions that capture those logical and semantic links you highlighted.
> > >
> > > Thank you again for your valuable feedback. We're glad our responses helped address your other concerns, and we look forward to further improving this work.

---

> > > ### Author Response · Authors · 2024-12-03
> > >
> > > Dear reviewer, as this is the last day for reviewer responses, we would like to follow up on our previous response and respectfully ask if you have remaining concerns about the contributions of our paper? We are more than happy to answer any further queries or concerns you may have, and thank you for your time and consideration during this discussion.

---

> ### Author Response · Authors · 2024-11-20
> **Response to weaknesses (evaluation metrics)**
>
> > 4. The proposed evaluation metrics may need more detailed analysis to show robustness. Current average weighting looks too simple ignoring the difference between specific models dealing with specific types of questions. Some penalty or reward terms may need to be considered.
>
> While average weighting may look simple, we believe that this may be a strength as a simple metric is easier to understand, implement, and interpret. We observe that this is a general approach also adopted by previous works “Replacing Judges with Juries: Evaluating LLM Generations with a Panel of Diverse Models”. On the other hand, we did not propose specific penalty or reward terms as they may introduce unexpected biases based on different models. Additionally, we believe the current approach can robustly aggregate the judge scores, as we observed a high Pearson correlation of 88.9% between the scores from our evaluation framework and human scoring (lines 343 - 347).
>
> > 5a. As the metrics are unexplored the results may not be comprehensive and reliable.
>
> To assess the quality of our evaluation metric, we conducted manual human scoring based on the same evaluation guide, observing a high Pearson correlation of 88.9% between our evaluation framework scores and the human annotator (lines 343 - 347). Thus, we believe that the evaluation metric is shown to be reliable and with high agreement to human preferences. This is also supported by previous works in model-based evaluation, such as “G-Eval: NLG Evaluation using GPT-4 with Better Human Alignment”, which demonstrates that similar model-based metrics can achieve high agreement with human evaluators.
> We also feel that the evaluation metric is comprehensive, as the guide shown in Figure 5 and Appendix A.3 considers aspects such as accurate information, comprehensiveness, relevance, and coherence. Could the reviewer suggest any specific aspects for the evaluation to be more comprehensive?
>
> > 5b. Lack of quantitative analysis show different domain, question types performance.
>
> As shown in Table 2 and Table 3, we do include the evaluation results on separate domains (academic, product, finance) and question categories (text, figure, table). Based on the results, we provided analysis into the specific areas of improvement for multimodal models, such as the lower performance on table-based questions (lines 428 - 450). For example in Table 3, Qwen2-VL achieved a correctness of 4.08, 3.83, 3.62 on text-based, figure-based, and table-based questions respectively.

---

> ### Author Response · Authors · 2024-11-20
> **Response to questions**
>
> > 1. Does your dataset consider the spanning-page answer setting?
>
> Currently we limit questions to single-page evidence to allow for a focused evaluation, establishing an important baseline before tackling multi-page reasoning. This approach enables the creation of a higher-quality dataset that is more straightforward to generate and evaluate at scale. While multi-page questions are an important future direction, the current setting remains challenging for open-source models and also reflects many real-world use cases. Overall, we believe the current setting addresses an existing gap in multimodal document understanding while laying the groundwork for more complex settings in future iterations.
>
> > 2. Is there a dataset structure parsing quality checking procedure?
>
> Yes, we leverage both automated and human verification to ensure that the multimodal content is correctly extracted and applicable to the question. For example, the verification includes whether the document content contains the specified category of text, figure, or table (194-203), and whether the question can be answered based on the extracted content. Thus if the content is not present due to document parsing failure or other issues, the sample would not be included in our dataset.
>
> > 3. Is there any analysis or comparison before and after human checking automatically generated QA pairs?
>
> As discussed in Section 2.2, we found that 80.1% of the generated questions satisfied the automated verification. Of these questions that passed automated verification, 80.9% also satisfied the human verification. Thus, we only retain 851 questions that satisfied both the automated and human verification (lines 203 - 205).
>
> > 4. Why does the number of document pages per document look wired, especially for academic papers? It might be too long for an academic paper.
>
> To collect multimodal documents in the academic domain, we sourced for thesis manuscripts in arXiv as they are longer in content. For the financial domain and product domain, we collect company annual reports and detailed product manuals respectively (lines 147-150).

---

> > ### Author Response · Authors · 2024-11-25
> > **[URGENT] your immediate attention is needed**
> >
> > Dear Reviewer rjZp,
> >
> > We hope this message finds you well. The discussion period is ending soon, I am writing to emphasize the importance of your review for our submission. Your score is significantly lower than the other reviewers, and we believe this discrepancy may indicate a misunderstanding or oversight.
> >
> > We have addressed all the concerns in our detailed rebuttal. We would appreciate your prompt attention to it. A thorough reassessment is crucial to ensure a fair evaluation.
> >
> > Your expertise is highly valued, and we trust that a reconsidered review will reflect the true merit of our work.
> >
> > Thank you for your immediate attention to this matter.
> >
> > Best regards, Authors

---

> > ### Author Response · Authors · 2024-11-26
> >
> > > 1b. And the author mentioned the in-depth but there is no supporting analysis and results to show why the dataset shows more in-depth.
> >
> > Dear reviewer, to further demonstrate the in-depth nature of our questions, we have manually categorized 100 questions in the table below, showing that they cover realistic aspects involving analytical reasoning, technical analysis, commonsense and domain knowledge, visual interpretation and mathematical reasoning, where many questions may involve multiple aspects. This analysis is also included in our revised version (lines 860-861).
> >
> >
> > | Category | Description | Proportion | Example Question |
> > |----------|-------------|------------|-------------------|
> > | Analytical Reasoning and Pattern Recognition | Questions about trends, comparisons, and implications (e.g., engagement trends, performance trends) | 49% | What is the total amount of financial liabilities at amortized cost for the year 2023, and how does it compare to the total amount for 2022? Consider the implications of any changes in these liabilities on the company's financial strategy. |
> > | Technical Analysis | Questions about specific technical details (e.g., UEFI BIOS, shutter speeds, X-sync speeds) and applications of technical concepts. | 37% | What potential issue could arise if you fail to follow the instruction to tighten the screws twice when installing the top cover, and why might this step be particularly important for a laptop? |
> > | Commonsense and Domain Knowledge | Questions requiring general knowledge or background knowledge in fields such as finance, cybersecurity, photography. | 46% | What are the key differences and potential advantages of using white-box analysis over machine learning for modeling the performance of configurable systems, as discussed by Velez et al. (2021)? |
> > | Visual Interpretation | Questions based on interpreting icons, diagrams, or charts. | 60% | Explain the functionalities of the different sections (a, b, c, d) in the LaserFactory design toolbar and discuss how each section contributes to the overall design and fabrication process. |
> > | Mathematical Reasoning | Questions involving mathematical concepts or calculation from data. | 17% | Calculate the percentage change in diluted net income per share attributable to common stockholders from fiscal year 2023 to fiscal year 2024. What factors likely contributed to this change? |

---

### Author Response · Authors · 2024-12-04
**Thanks to all reviewers and general summary**

Dear reviewers and chairs,

We are delighted to note that reviewer t1TM and rjZp felt their concerns were addressed following our clarifications and revisions, reviewer rjZp increased their score, and reviewer Jfm1 gave a very positive feedback with no major weaknesses. In general, the reviewers noted our challenging research problem (rjZp), strengths of the dataset and evaluation framework for multimodal long documents (rjZp, Jfm1), and novel retrieval-aware tuning approach (jfm1, t1TM). Reviewers t1TM and Jfm1 also noted the novelty of our dataset, which has much longer documents and open-ended answers compared to prior works.

We would like to highlight our key contributions and summarize the enhancements we have made to our submission following the insightful suggestions of the reviewers. Our unique contributions are:

1. A comprehensive baseline for in-depth multimodal understanding of individual pages in long documents, isolating text-based, figure-based, and table-based questions across diverse domains.
2. A focus on open-ended questions requiring longer answers, incorporating aspects such as analytical reasoning, technical analysis, commonsense knowledge, and mathematical reasoning.
3. An automated and reliable evaluation framework for open-ended multimodal document question answering.
4. Detailed analyses of leading models, revealing multimodal biases and practical challenges in leveraging retrieval methods.
5. A novel retrieval-aware tuning approach that improves model performance by up to 4.6%, applicable to many long multimodal document applications.

Following the insightful suggestions of the reviewers, we have made the following enhancements to revise our submission:

1. Detailed breakdown of question types and examples, covering analytical reasoning, commonsense knowledge, and more (t1TM, rjZp).
2. More detailed clarification of text-only vs multimodal results (t1TM).
3. Discussion of DocBench in related works section (t1TM).
4. Comparison showing that our questions and answers are much longer than previous multimodal document benchmarks (rjZp).

Thank you all once again for the valuable feedback and for facilitating the improvement of our work.

Best regards, Authors

---

### Meta-Review · Area_Chair_WzpZ · 2024-12-23

**Metareview:**

**Summary:**
The paper introduces M-LongDoc, a benchmark for evaluating multimodal models on lengthy documents containing text, figures, and tables, with open-ended questions requiring deep analysis. It proposes a retrieval-aware tuning framework to improve model performance by training with both relevant and distracting content. The framework show reasonable improvement in correctness, addressing multimodal biases and challenges in long document understanding​

**Strength:**
- It tackles a challenging and necessary research problem with applications across various domains.
- It introduces a new dataset, M-LongDoc, for multimodal long document understanding, having lengthy pages per document with open-ended questions.
- It provides valuable resources, including a new evaluation dataset, training dataset, evaluation framework, and multimodal RAG-QA model.

**Weakness:**
- Questions focus on specific pages, ignoring answers spanning multiple pages; lacks evidence to support claims of in-depth analysis.
- The average weighting approach is overly simplistic, ignoring model-specific differences and lacking penalty/reward mechanisms.
- Evaluation depends on proprietary LLMs, introducing variability due to differing checkpoints or versions.
- Missing discussion of some related works and insufficient differentiation from them.

**Additional Comments On Reviewer Discussion:**

This paper received three reviews, two positive and one negative. While the paper demonstrates several merits, there is room for improvement, such as addressing the limitation of answers spanning multiple pages and the reliance on synthetic data generation. So, this paper slightly falls short of the acceptance bar for the ICLR conference, especially when compared to the higher ratings and contributions of other submissions.

---

### Decision · Program_Chairs · 2025-01-22

Reject